# A Straightforward and Efficient Instance-Aware Curved Text Detector

**DOI:** 10.3390/s21061945

**Published:** 2021-03-10

**Authors:** Fan Zhao, Sidi Shao, Lin Zhang, Zhiquan Wen

**Affiliations:** Department of Information Science, Xi’an University of Technology, Xi’an 710054, China; 2180820019@stu.xaut.edu.cn (S.S.); 2180821086@stu.xaut.edu.cn (L.Z.); 2180820012@stu.xaut.edu.cn (Z.W.)

**Keywords:** text detection, convolutional neural networks, article swarm optimization, curved text

## Abstract

A challenging aspect of scene text detection is to handle curved texts. In order to avoid the tedious manual annotations for training curve text detector, and to overcome the limitation of regression-based text detectors to irregular text, we introduce straightforward and efficient instance-aware curved scene text detector, namely, look more than twice (LOMT), which makes the regression-based text detection results gradually change from loosely bounded box to compact polygon. LOMT mainly composes of curve text shape approximation module and component merging network. The shape approximation module uses a particle swarm optimization-based text shape approximation method (called PSO-TSA) to fine-tune the quadrilateral text detection results to fit the curved text. The component merging network merges incomplete text sub-parts of text instances into more complete polygon through instance awareness, called ICMN. Experiments on five text datasets demonstrate that our method not only achieves excellent performance but also has relatively high speed. Ablation experiments show that PSO-TSA can solve the text’s shape optimization problem efficiently, and ICMN has a satisfactory merger effect.

## 1. Introduction

As a crucial premise of text recognition, scene text detection has largely attracted the attention of many academics and industrial researchers, and many promising results have been achieved in recent decades. However, due to existing large differences in size, aspect ratio, direction and shape, as well as the presence of distortion and occlusion, success detection of scene text is still a very challenging task. To deal with these challenges, traditional text detection pipelines [1,2] usually focused on two subtasks: text detection and non-text removal, however, they are limited by hand-crafted features and usually involve heavy post-processing.

In recent years, with the renaissance of convolutional neural networks (CNNs), many deep learning-based methods [3,4,5,6,7,8,9,10,11,12,13,14,15,16,17,18,19,20,21,22,23,24,25,26,27,28,29,30,31,32,33,34] have achieved remarkable achievements in text detection, and these methods can be divided into top-down and bottom-up methods. The top-down methods [3,4,5,6,7,8,9,10,11,12,13,14,15,16,17,18,19,20], also commonly referred to as regression-based methods, usually adopt popular object detection pipelines to first detect text on the block level and then break a block into the word or line level if necessary. However, because of the structural limitations of the corresponding CNN models, these methods cannot efficiently handle long text and arbitrarily shaped texts. The bottom-up methods [21,22,23,24,25,26,27,28,29,30,31,32,33,34] first detect text components with a CNN and then group these components into text instances. According to the granularity of the text components, the bottom-up methods are mainly divided into two categories: pixel-level methods and part-level methods. Borrowing from the idea of semantic segmentation, pixel-based methods [21,22,23,24,25,26,27,28,29] produce a text saliency map for text detection by employing an FCN [30] to perform pixel-level text/nontext prediction. By regarding a text region as a group of text components, part-level methods [31,32,33,34] usually first detect individual characters and then group them into words or text lines. Compared with the top-down methods, the bottom-up methods have more flexibility and better detection performance. Unfortunately, however, the training model requires a large number of sample annotations and complex network design, which limits their generalization.

For taking advantage of the high efficiency of top-down methods while avoiding the bottom-up method for large-scale labeling of samples, we introduce an efficient and straightforward curved text detection method, namely, look more than twice (LOMT), which progressively localizes a complete text instance multiple times without heavy learning. First, the text proposals are located in the original image by adopting a direct regression network, and then the candidate characters are extracted from each text proposal by using maximally stable extremal regions (MSER). Next, a particle swarm optimization-based text shape approximator (PSO-TSA) gradually approximates an arbitrary text shape by a specific PSO technique. Finally, one complete text instance is generated by merging two adjacent or intersecting text proposals by an instance-aware component merging network (ICMN).

Our main contributions can be summarized as follows:(1)We propose a text detector, which can accurately locate curved text appearing in the scene through several consecutive straightforward and effective steps.(2)To fine-tune the result of the regression-based text detector, we propose a particle swarm optimization-based text shape approximator called PSO-TSA, which can quickly approach text shapes without heavy pre-training or pre-learning in advance.(3)To improve the text instance completeness, an instance-aware component merging network (ICMN) is designed to merge adjacent text subparts, which can be flexibly adapted to text detection results of any shape.(4)Although the entire pipeline is not an end-to-end mechanism, the experiments on five text benchmark datasets show that our method not only achieves excellent harmonic mean (H-mean) performance but also has relatively high speed.

The rest of this paper is organized as follows. We briefly review some related works on scene text detection in Section 2, followed by describing the details of the proposed method in Section 3. Then, we present experimental results in Section 4. Finally, we conclude and give some perspectives on future work in Section 5.

## 2. Related Works

Recently, numerous approaches have been proposed to address different challenges in text detection, such as low resolution, perspective distortion, low contrast, complex background, multi-lingual, multi-view, multi-direction, and arbitrary shape and size. In early works, connected component analysis (CCA) was mainly utilized to detect text in scene images by first extracting candidate text fragments and then filtering non-text candidates. Representative connected-component-based approaches such as the stroke width transform (SWT) [1] and MSER [2] have achieved outstanding performance on various test datasets, particularly for well-focused texts, e.g., ICDAR13 [35]. However, these methods fall short for the more challenging ICDAR15 [36] and MSRA-TD500 [37] due to the limitation of hand-crafted features. With the advance of CNNs, different CNN models have been exploited for scene text detection tasks. In general, CNN-based methods can be roughly formulated into two categories: top-down methods and bottom-up methods.

Top-down methods first detect text on the block level and break a block into the word or line level if necessary, and these methods are often based on general object detection frameworks [3,4,5,6,7,8,9,10,11,12,13,14,15,16,17,18,19,20]. The Faster Region-based CNN (R-CNN) [3] is the most representative and accurate generic object detection method and has also achieved promising results on text detection tasks [4]. TextBoxes++ [5] adopts irregular 1 × 5 convolutional filters instead of the standard 3 × 3 filters and leverages recognition results to refine the detection results. ITN [6], E2E-MLT [11] and FOTS [18] are end-to-end text instance networks. Liu et al. applied feature pyramid networks (FPNs) in the CNN part to extract features of multi-scales, and used the bidirectional long short-term memory (Bi-LSTM) network to generate text proposals [7]. By connecting a region proposal network (RPN) and a regression module (TLOC), Liu et al. [8] proposed a method named the curve text detector (CTD) to detect curved texts. EAST [9], deep regression [20], DeconvNet [10] and E2E-MLT [11] all directly use a fully convolutional network (FCN) to predict the score map for each pixel. Inspired by EAST, Zhang et al. [12] presented a text detector named look more than once (LOMO) that progressively localized text multiple times. SegLink++ [13] detects dense and arbitrarily shaped scene text by instance-aware component grouping. By combining deep learning and histogram-oriented gradient features, NguyenVan et al. [14] designed a pooling-based scene text proposal technique and integrated it into an end-to-end scene text reading system. He et al. [15] proposed multi-scale scene text detection with a scale-based RPN. Zhu et al. [16] presented a shape robust detector with coordinate regression. Wang et al. [17] proposed a progressive scale expansion network (PSENet) to detect text instances with arbitrary shapes. Wang et al. [19] proposed a text region proposal network (text-RPN) and verified and refined it through a refinement network.

The above state-of-the-art regression-based methods have achieved promising results on standard benchmarks. However, most of these methods need a comprehensive design with complex anchors and cumbersome multistage network structure, and suboptimal performance is achieved by exhaustive tuning. Unsatisfactory results often occur when dealing with long text.

Bottom-up methods usually first detect text at the pixel or part level and then segment these components into desired word- or line-level text instances. Borrowing from the idea of semantic segmentation, pixel-based methods [21,22,23,24,25,26,27,28,29] produce a text saliency map by employing an FCN [30] to perform pixel-level text/non-text prediction. Wu et al. [21] proposed self-organized text detection with minimal post-processing via border learning. He et al. [22] presented an end-to-end scene text detection by combining a multiscale FCN with a cascade-style instance segmentation. PixelLink [23] generates an 8-direction margin to separate text lines. Xue et al. [24] exploited bootstrapping and text border semantic techniques for long text localization. Zhang et al. [25] first used an FCN to extract text blocks and then search for character candidates from these blocks with MSER [2]. By fusing the proposed generator technology with the FCN, a framework named FAST [26] was proposed to reduce the number of text proposals while maintaining a high recall in scene text images. Xu et al. [27] presented a text detector named TextField, which directly regresses the direction field. Pixel-level methods mostly need complex post-processing to extract accurate bounding boxes of words or text lines. Liu et al. propose the Mask Tightness Text Detector (Mask TTD) [29], which uses a tightness prior and text frontier learning to enhance pixel-wise masks prediction.

By regarding a text region as a group of text components, part-level methods [30,31,32,33,34] usually first detected components or characters and then grouped them into words or text lines. SPC-Net [30] uses an instance segmentation framework and context information to detect text in arbitrary shapes while suppressing false positives. Lyu et al. [32] combined position-sensitive segmentation with corner detection to generate text instances. Long et al. [33] used ordered disks to represent curved text. Baek et al. [34] developed a framework, referred to as character region awareness for text detection (CRAFT), which explores each character and the affinity between characters.

This work is inspired by the idea of LOMO [12], which progressively localizes text multiple times. However, the detection results of LOMO are multiple subsections of a word or text line instead of a single complete text instance, which leads to a certain degree of incompleteness of semantics. Similar to LOMO, by combining the backbone of ResNet50 [38] with FPN [39], our method also integrates the advantages of top-down methods and bottom-up methods. In order to overcome the limitation of regression-based detector to locate arbitrary shape text and to make the located text box to fully contain text while fitting its out-line, we first propose a straightforward and efficient text shape approximation method called PSO-TSA, which is based on particle swarm optimization method. Without prior training and heuristic parameters, PSO-TSA achieves very competitive performance on curved and non-curved datasets. Moreover, a lightweight ICMN is proposed to merge two adjacent or intersecting text subsections in a text line, which can further improve the completeness of a text instance. Compared with most deep learning algorithms, the advantages of our method are very simple and straightforward, PSO-TSA is general to any data set and ICMN can be readily plugged into any CNN-based detector.

## 3. The Proposed Method

### 3.1. Background

A challenging aspect of scene text detection is to handle curved text which is common in natural scenes. The regression-based methods have achieved promising results on standard benchmarks, and the efficiency is also very high. Using regression-based method for text proposal detection is a premise of our method. Here, we choose the classic EAST detector [9] to detect the text proposals appearing in the scene.

By inputting the text image to EAST detector [9], a dense prediction channel of text/non-text is outputted to indicate the pixel-wise confidence of being text or not. The dense proposals from the network are then filtered based on the text confidence score rp with the threshold value 0.9. For further processing in the next step, the pixel proposals with rp≥0.9 are labeled as white pixels, and the rest are labeled as black pixels, which are also considered text pixels and non-text pixels, respectively. The image formed by these white/black pixels is called a text confidence map, marked as the symbol $I_{m}$. Finally, locality-aware NMS [9] is performed on the filtered proposals to generate candidate quadrilateral text proposals, which are collected into a text proposal set $R=\left\{r_{t}\left|{}_{t\in \left[1,T\right]}\right.\right\}$. Here, $r_{t}$ is the t−th text proposal, and T is the total number of text proposals.

Usually, regression-based text detectors fall quite short of detecting extremely long curved text. In order to overcome the limitation of regression-based detector to irregular text, we introduce an efficient and straightforward mechanism called PSO-TSA, which makes the regression-based detection result gradually change from loosely bounded rectangle box to compact polygon.

In the PSO-TSA method, we need to fine-tune the text proposal detected by EAST. Character extraction is another prerequisite of our method. The accuracy of character extraction in the text proposal largely determines the performance of our method. In order to avoid a large number of sample annotations and complicated network designs, we adopt a simple traditional character extraction method like MSER [2] instead of a deep framework like CRAFT. The reason for choosing MSER is that the MSER algorithm assumes that text components have a similar homogeneous background and foreground and thus are stable with respect to a range of thresholds. Here, an extracted connected component is also considered a candidate character. The candidate characters extracted from the t−th text proposal $r_{t}$ are then combined into a set $C_{t}$, expressed as $C_{t}=\left\{c_{m}^{t}\left|m\in \left[1,M\right]\right.\right\}$, in which $c_{m}^{t}$ is the m−th character in t−th text proposal and M is the total number of candidate characters. It should be noted here that a detected connected component does not refer to an individual character in a strict sense; it may be a single character or consecutive multiple characters in a text proposal.

### 3.2. LOMT

As illustrated in Figure 1, the pipeline of our approach consists of text proposal detection, connected components (CCs) extraction, shape approximation and component merging, the first two are prerequisites, and the last two are the main strategies of our algorithm.

#### 3.2.1. PSO-TSA

Inspired by the sociological behavior associated with birds flocking, Kennedy and Eberhart proposed a PSO method [40] to solve a global optimization problem. Each individual in the swarm is called a particle, which represents a candidate position or solution to the problem. By constantly adjusting their positions and speeds with shared information or experience, particles fly through the search space influenced by two factors: one is the individual’s best position ever found (pbest); the other is the group’s best position (gbest). The PSO algorithm possesses the advantages of simplified computing, rather quick convergence speed, global optimization performance and fewer control parameters, so we apply an improved PSO in this paper to solve the optimal approximation problem of an arbitrary text shape. What needs to be pointed out here is that in addition to the position and velocity properties of the particles, we introduce isometric information into the particle in order to better use the improved PSO algorithm to approximate the shape of the text. To the best of our knowledge, this is the first time that the PSO algorithm has been used in curved text detection. A vivid example is shown in Figure 2 to explain the procedure of the proposed PSO-TSA algorithm, in which the text proposals and candidate characters are extracted from EAST and MSER algorithm, respectively. First, a curve is fitted with all the center points of the candidate characters, which is called the fitted character centerline. Because the accuracy of character detection by MSER algorithm is normally not high enough, this fitted curve is most likely not the real centerline of the curved text, which is also a problem that is usually faced by bottom-up algorithms. In response, points are evenly sampled on the fitted character centerline, which serve as the initial points of the particle, and also as the center point of the sampling neighborhood of the particle’s corresponding dimension position. Here, the sampling neighborhood is also the region where the particle position changes, and particle positions in all dimensions are fitted to generate the approximated centerline of the text. Because the particle’s initial position points are sampled on the fitted character centerline, the first approximated centerline is the same as the fitted character centerline. However, as the particle’s position changes when the particle swarm flies in the sampling neighborhood, the approximated centerline will be farther away from the fitted character centerline and closer to the real centerline of the text such that it eventually coincides with the real text centerline. The polygon, connected in series by equidistant control points moving in the direction perpendicular to the approximated centerline, is the approximate polygon for curved text by PSO-TSA algorithm. By combining the shared foraging flight of the particle swarm and the continuous adjustment of the equidistant points of the corresponding particles according to their distance change, the global optimal particle position gradually approaches the true centerline of the text; in the same instant, the polygon gradually approaches the text shape, and finally, both optima are reached.

The specific initialization of particles is described in detail in Algorithm 1. First, we define a particle swarm X=X1,X2,⋯,XI, in which Xi represents the i−th particle, i∈1,I, and I is the total number of particles in particle swarm X. Each particle is represented by its position, velocity and distance between its position and equidistant points, namely, Xi=Xi.p, Xi.v, Xi.dT. Before initializing the particles, we first need to set the particles’ range of activity. Assume that N points are uniformly sampled on the character centerline, which are expressed as P1,P2,⋯,PN. The spatial neighborhood Nr1×r1Pn with center point Pn and radius r1 is the variation range of the n-th dimension position xin of Xi.p for i∈1,I & n∈1,N. That is, the position of particle Xi is expressed as Xi.p=xi1,xi2,⋯xiNT, in which xin∈Nr1×r1Pn. xin is initialized as a point Pns sampled in Nr1×r1Pn, whose x and y coordinates are equal to Pns. x and Pns. y, respectively. The n-th dimensional velocity vin of Xi.v is initialized to 0, 0T. The distance variable din between the n-th position point xin and its two equidistant points is initialized with a value uniformly sampled in the interval d1, d2.
**Algorithm 1.** Particle Swarm Initialization.1.  Define particle swarm X=X1,⋯,Xi, ⋯, XI, in which Xi⋅p=xi1,⋯,xin,⋯,xiNT, Xi.v=vi1,⋯vin,⋯,viNT and Xi.d=di1,di2,⋯diNT, respectively.2.  Construct N spatial neighborhoods, each denoted as Nr1×r1Pn, n∈1, N.3.  Set the range of equidistant values of particle position points as d1,d2.4.  ***for*** each Xi∈X
***do***5.   Randomly sample a point Pns in each Nr1×r1Pn, and all sampled points form a point set Ps=P1s, ⋯,Pns, ⋯,PNs.6.   Randomly sample N values in interval d1,d2 to form a equidistant set DS=d1s, ⋯,dns, ⋯,dNs.7.   ***for*** each n∈1,N
***do***8.    xin=Pns⋅x,  Pns⋅yT, vin=0,0T and din=dns.9.   ***end for***10. ***end for***

When the number of iterations k does not reach the maximum number of iterations K or the error criterion Δε does not reach the minimum $\varepsilon_{\min}$, the proposed PSO-TSA algorithm is implemented iteratively to approximate the text shape. Each cyclical process mainly includes the following three steps:

**Step 1:** Two equidistant points $y_{i}^{n,1}$ and $y_{i}^{n,2}$ are calculated in the normal direction at $X_{i}{}^{’}s$
n−th position point $x_{i}^{n}$ according to Equations (1) and (2), respectively.
(1)$$\left\{\begin{array}{c} y_{i}^{n,1}\cdot x=x_{i}^{n}-d_{i}^{n}\times \sin\theta_{n}\\ y_{i}^{n,1}\cdot y=x_{i}^{n}+d_{i}^{n}\times \cos\theta_{n} \end{array}\right.$$
(2)$$\left\{\begin{array}{c} y_{i}^{n,2}\cdot x=x_{i}^{n}+d_{i}^{n}\times \sin\theta_{n}\\ y_{i}^{n,2}\cdot y=x_{i}^{n}-d_{i}^{n}\times \cos\theta_{n} \end{array}\right.$$

Here, $\theta_{n}$ is the angle between the tangent line at point $x_{i}^{n}$ and the horizontal positive half-axis, that is, $\theta_{n}=\arctan\left(k_{n}\right)$. $k_{n}$ is the tangent slope at the n-th point $x_{i}^{n}$ of particle $X_{i}$. The isometric points of all position points are connected head to tail to form the corresponding polygon $S_{i}$ of particle $X_{i}$. 

**Step 2:** The valuating indicator $f_{x_{i}}$ of particle $X_{i}$ is calculated as
(3)$$f_{X_{i}}=0.5\times \frac{A_{S_{i}}}{A_{r_{t}}}+0.5\times \frac{N_{S_{i}}^{c}}{N_{r_{t}}}$$

The fitting value $f_{X_{i}}$ is expressed as the sum of two parts: one corresponds to the ratio of the text aggregation degree in $S_{i}$ and $r_{t}$, and the other corresponds to the ratio of the number of characters in $S_{i}$ and $r_{t}$. Each part has a weight of 0.5. Here, $A_{S_{i}}$ and $A_{r_{t}}$ are the text aggregation degree of $S_{i}$ and $r_{t}$, respectively. $N_{S_{i}}^{C}$ and $N_{r_{t}}$ is the number of characters in $S_{i}$ and $r_{t}$. Here the variable $N_{r_{t}}$ takes the value M. Given the text confidence map $I_{m}$, which is obtained in Section 3.1, the text aggregation degree $A_{S_{i}}$ is defined as the ratio of the number of white pixels in the polygon $S_{i}$ within $I_{m}$ to the area of $S_{i}$, and $A_{r_{t}}$ is the ratio of the number of white pixels in the text proposal $r_{t}$ to the area of $r_{t}$. 

The reason for considering the parameter ratio rather than the parameter itself of each part is that doing so can ensure the completeness of the text instance while ensuring the compactness.

**Step 3:** After k−th single loop ends, the individual particle optima $Pbest_{i}$ and the particle swarm global optimum $Gbest^{k}$ are found from the history, and each particle’s velocity, position and distance are updated according to Equations (4)–(6), respectively, in which $c_{1},\;c_{2}$ represent the acceleration constants, $\mu_{1},\;\mu_{2}$ are uniformly distributed random numbers and Δδ is white Gaussian noise.
(4)$$X_{i}^{k}.v=X_{i}^{k-1}.v+c_{1}\times \mu_{1}\times (Pbest_{i}-X_{i}^{k}.p)+c_{2}\times \mu_{2}\times (Gbest^{k}-X_{i}^{k}.p)$$
(5)$$X_{i}^{k}.p=X_{i}^{k-1}.p+X_{i}^{k}.v$$
(6)$$X_{i}^{k}.d=X_{i}^{k-1}.d+\Delta \delta \;$$

When the optimization is completed, the optimal polygon will be obtained, which is composed of 14 equidistant points connected in series after the PSO-TSA optimal approximation.

#### 3.2.2. ICMN

Even if the PSO-TSA algorithm is used to revise the EAST detection result from quad-rilateral to polygon, the approximate polygon may not be a complete text instance but only a part of the text instance. To avoid incomplete scene text detection when processing long words or text lines, it is necessary to merge adjacent or intersecting text components that belong to the same text instance. Therefore, ICMN is proposed to solve the instance incomplete detection problem. The intuition behind the ICMN is that humans can easily judge whether two separate text components belong to a complete text instance and naturally perceive its shape without examining the entire instance; similarly, a specially designed neural network might also be able to infer the boundary points of a text instance.

The specific architecture of the ICMN is shown in Figure 3, which consists of an input layer, a pooling layer, two fully connected layers (FC) and an output layer. The input layer first accepts the boundary points of two adjacent polygons as individual features, both of which have a size of 1×28. In the pooling layer, these two features from the input are simply concatenated together to build a 1×56-sized snippet feature. There are two fully connected layers in the ICMN, the sizes of which are 56×48 and 48×30. Using fully connected layers in the ICMN reduces the dimensions of the pooled feature to the desired size of 1×30. The final output consists of two sibling parts: the first one (with 1×2−sized) outputs two confidence scores indicating whether the two input components belong to the same text instance or not, and the second one (with 1×28−sized) outputs 14 regression boundary points of the instance, half of which are upper boundary points and half of which are lower boundary points. Because each point has x and y coordinates, the ICMN outputs a total of 28 corresponding point coordinates. The salient aspects of the ICMN are that it can predict whether two components merge and can refine the text instance boundary points by spatial coordinate regression. 

For training the ICMN, we assign a binary class label (whether it belongs to an instance) to each text snippet. A positive label is assigned to a snippet if (i) two subsections in it belong to an instance or (ii) the Intersection over Union (IoU) is greater than or equal to 0.3. We design a multitask loss L to jointly train classification and coordinate regression.
(7)$$L=L_{cls}+\lambda L_{reg}$$

Here, $L_{cls}$ is the loss for merging/non-merging classification, which is a standard softmax loss; $L_{reg}$ is the loss for spatial coordinate regression; and λ is a hyperparameter used to control over fitting. The regression loss is
(8)$$L_{reg}=\frac{1}{N_{pos}}\sum_{i=1}^{N_{b}}\left(l_{i}^{*}\times \frac{1}{14}\sum_{j=1}^{14}{\left\|\left(P_{j}^{i}.x-P_{j}^{i*}.x\right)+\left(P_{j}^{i}.y-P_{j}^{i*}.y\right)\right\|}_{2}^{2}\right)$$
where $L_{2}$ distance is adopted; $l_{i}^{*}$ is the label, which is 1 for positive samples and 0 for negative samples; $N_{pos}$ is the number of positive samples in a mini-batch; and $N_{b}$ is the number of all samples in a mini-batch. Here, $P_{j}^{i}.x$ and $P_{j}^{i}.y$ denote the x and y coordinates of the predicted j−th point $P_{j}^{i}$ of the i−th sample, respectively, and $P_{j}^{i*}.x$ and $P_{j}^{i*}.y$ denote the x and y coordinates of the j−th ground truth point of the i−th sample, respectively. Before training, we obtain the real ground truth by the method proposed in [8]. The regression loss is calculated only for positive samples. To assist LOMT in dealing with long curved text, a small and light ICMN is used in conjunction with PSO-TSA. Two adjacent or intersecting polygons may be subparts of the same text instance, so they are input to the ICMN to decide whether to merge them into a more complete text instance. The input of the ICMN is 28 points formed by concatenating each pair of 14 points of adjacent polygons, and the outputs of the network are the merging flag of the two parts and the regressed 14 points of the more complete text polygon. After a series of merging steps, each complete text instance appearing in the scene can be detected.

## 4. Experiment and Discussion

### 4.1. Datasets and Implementation Details

#### 4.1.1. Datasets

ICDAR2015 (IC15) [36]: IC15 was built for Challenge 4 of the ICDAR-2015 Robust Reading Competition and consists of 1000 training images and 500 test images. The images are acquired using Google Glass, and the text accidentally appears in the scene. The ground truth is annotated by a word-level quadrangle.

ICDAR2017 (IC17) [41]: IC17 contains 7200 training images, 1800 validation images, and 9000 test images with text in 9 languages for multilingual scene text detection. Similar to IC15, the text regions in IC17 are also annotated by the 4 vertices of quadrilaterals.

MSRA-TD500 (TD500) [37]: TD500 contains 500 natural images, which are split into 300 training images and 200 test images, collected both indoors and outdoors using a pocket camera. The images contain English and Chinese scripts. This dataset is mainly used for multidirectional text detection, in which text regions are annotated by rotated rectangles.

SCUT-CTW1500 (CTW1500) [8]: CTW1500 consists of 1000 training and 500 test images. Every image has curved text instances, which are annotated by polygons with 14 vertices. The annotation is given at the text-line level such that a complete sentence is annotated as a single polygon.

Total-Text (Total-Text) [10]: Total-Text contains 1555 scene images, which are divided into 1255 training images and 300 test images. This dataset mainly provides curved texts, which are annotated at the word level by polygon-shaped bounding boxes.

#### 4.1.2. Implementation Details

In this section, we carry out experiments on curved and non-curved datasets to test our method. In actual experiments, while maintaining the aspect ratio, the longer sides of the images in IC15, IC17, and TD500 are resized to 768, 1024, and 768, respectively, the longer sides of the images within Total-Text and CTW1500 are resized to 1280 and 1024, respectively. All experiments are performed with a single image resolution. The results are evaluated using the standard PASCAL VOC protocol [42], which uses a 0.5 intersection-over union (IoU) threshold to determine true and false positives.

For training the text proposal detector, a union of the IC15 and IC17 dataset is used. The network is trained using an improved adaptive moments (ADAM) [43] optimizer. To speed up learning, we uniformly sample 512 × 512 crops from images to form a mini-batch of size 24. The learning rate of ADAM starts from $1\times 10^{-3}$, decays to one-tenth every 25,000 mini-batches, and stops at $1\times 10^{-5}$. The network is trained until the performance stops improving. 

When executing the PSO-TSA model, the parameters are set as follows. The particle swarm population consists of I particles moving in the N-dimensional search space, in which I and N are set as 20 and 7, respectively. The radius of the sampling neighborhood, r1, is a certain proportion of the maximum of the average width and height of characters, which is set to 0.25 in the experiment. The distance range of the particle’s position point and its two equidistant points is $\left[d_{1},\;d_{2}\right]$, in which $d_{1}=1.5\times r1$ and $d_{2}=2.5\times r2$. The maximum number of iteration K and the minimum error $\varepsilon_{min}$ are set 20 and $1\times 10^{-3}$ respectively. The particle’s acceleration constants $C_{1}$ and $C_{2}$ are the same, both of which are set to 1.2, and both $u_{1}$ and $u_{2}$ are random numbers in (0, 1). 

In the objective function of ICMN, the hyper-parameter λ was set as 1.5. During training of the ICMN network model, the negative-to-positive sample ratio was set to 10 in a mini-batch, the learning rate was set to 0.005, the batch size was 128, and the number of iterations was set to 30,000. In text detection process, when the IoU of two polygons is greater than or equal to 0.3, the two polygons are input into ICMN network for further merging.

### 4.2. Comparison with State-of-the-Arts

#### 4.2.1. Experiments on Multi-Oriented Text (IC15, IC17 and TD500)

We first conduct experiments on three multi-oriented text datasets, IC15, IC17, and TD500, to demonstrate that the proposed method performs well for multi-oriented scene text detection. For IC15, annotations are provided at the word level, so the test experiments are performed at the word level. For the IC17 and TD500 datasets, annotations are provided at the line level, without a post-processing step applied to generate word boxes, similar to the method in [34]. We evaluate the performance of the proposed method directly at the line level. To make a fair comparison with end-to-end methods [11,18], we include their detection-only results by referring to the original papers. We also compare the proposed method with other state-of-the-art methods, including five recent multi-oriented text detectors, EAST [9], TextBoxes++ [5], FTPN [7] and the methods in [25,32] and five arbitrarily shaped text detectors: TextSnake [33], PixelLink [23], PSENET-1s [17], CRAFT [34] and ICG (SegLink++) [13]. The comparison is given in Table 1. As depicted in Table 1, LOMT achieves 85.7%, 74.6%, and 82.7% in terms of the harmonic mean (H-mean) on IC15, IC17, and TD500, respectively, outperforming most competitors.

IC15 is a dataset of complicated backgrounds, and the text size is small. As shown in the fourth to sixth columns of Table 1, the H-mean performance of LOMT is better than all other competitors, except that it is slightly lower than the CRAFT method. In addition, we demonstrate some test examples in Figure 4a, and LOMT can accurately locate text instances in different directions and different sizes.

IC17 is a large-scale multilingual text dataset. We next verify the ability of our method to detect multilingual text for IC17. As shown in columns 7 to 9 of Table 1, our method outperforms all other competing methods in terms of H-mean. Some qualitative detection results on the IC17 dataset are given in Figure 4b, and the results show that our LOMT method can accurately detect multilingual text such as English, Chinese and Korean in different directions and sizes.

On MSRA-TD500, we evaluate the performance of LOMT for detecting long and multilingual text lines. As shown in the results of columns 10 to 12 of Table 1, our method has certain advantages in the detection of long texts, and the H-mean is only 0.2% lower than CRAFT [34]. In addition, we demonstrate some test examples in Figure 4c, and our method can accurately locate long text instances with various orientations.

#### 4.2.2. Experiments on Two Curve Text Datasets (Total-Text and CTW1500)

To test the ability of LOMT for the detection of curved text, we evaluate the detection results on both the Total-Text and CTW1500 datasets, which mainly contain curved text.

We first conduct text detection experiments on the Total-Text dataset. Total-Text was constructed in late 2017, and curved text was collected from various scenes, including text-like and low-contrast background scenes. Most images in this dataset contain at least one curved text, which is annotated in word-level granularity using a strictly 14-point polygon. To compare LOMT with other algorithms under the same conditions, the number of uniformly sampled points on the character centerline is set to 7 so that the polygon with the same 14 vertices as the ground truth. Columns 3 to 5 of Table 2 show the results of different methods on Total-Text. As shown in Table 2, the proposed method achieves 88.2%, 79.6%, and 83.7% in precision, recall and H-mean on Total-Text, respectively. The results show that our method outperformed recent state-of-the-art methods on Total-Text. Especially, compared with the text detector EAST, the H-mean is improved by 41.7%, and our method is slightly better than the excellent CRAFT algorithm. Examples of the detection results of the proposed method are illustrated in Figure 5. From the results, we can find that the proposed method can work well for text of any shape and length. In addition, we show some test examples of different methods on Total-Text in Figure 6, where the results in columns (a), (b) and (c) correspond to CTD + TLOC [8], CRAFT [34] and the proposed LOMT method, respectively. CTD + TLOC [8] detects a false text in the first row in Figure 6a, and all detection results are inaccurate. The detection results of CRAFT [34] shown in Figure 6b are relatively ideal, but the parts of a single character, such as the letter “C”, are not completely contained in the detected text box. Compared to other methods, the proposed method has better completeness of the text instances and smoother boundaries of the polygons.

We continue to evaluate the results of LOMT on CTW1500, whose annotation is given at the text-line level such that a complete text instance is annotated as a single 14-point polygon. A comparison with the quantitative evaluations of other methods is depicted in Table 3, indicating that H-mean performance of the proposed LOMT method is 22.1% higher than the EAST [9] algorithm. Compared with CTD + TLOC [8], TextSnake [33], PSENet-1s [17] and Mask-TTD [29], H-mean improvement is 9.1%, 6.9%, 4.5% and 3.1%, respectively. The proposed LOMT is slightly better in H-mean performance than ICG (SegLink++) [13] and the method in [19] and is only 1.0% lower than CRAFT [34]. The results demonstrate that the proposed method can handle arbitrarily shaped line-level text detection. Some qualitative results are shown in Figure 7. The results show that LOMT can handle long text detection well.

#### 4.2.3. Speed and Ablation Analysis

All the experiments are conducted on a PC using a single NIVIDA GeForce GTX 1080 graphics card with the Pascal architecture and an Intel(R) Core(TM) i7-6700K @ 4.00 GHZ CPU. The speed of PSO-TSA and ICMN are 20.0 FPS and 30.9 FPS, respectively, which can basically achieve real-time running. Although the H-mean performance of the CRAFT algorithm is currently the most competitive compared with other state of the arts, its average frame rate is only 3.3.

The proposed method consists of four modules: text proposal detection, candidate character extraction using MSER, text shape approximations by PSO-TSA, and text subsection merging via ICMN. The first two modules are the prerequisites for PSO- TSA. Without the extracted characters, the PSO-TSA module cannot be executed independently. That is, the contribution of the character extraction module is included in the performance improvement of PSO-TSA. Therefore, we need to verify only the effectiveness of the three modules, namely, text proposal detection, PSO-TSA and ICMN. Table 3 shows the ablation experiment results on two curved text benchmarks. From Table 3, for the CTW1500 and Total-Text datasets, compared to the text proposal detection module, the improvements in the H-mean performance of the PSO-TSA module are 10.8% and 39.7%, respectively. Furthermore, compared with the PSO-TSA module without ICMN, the gain of the H-mean performance of LOMT is 10.3% and 2.0%. In particular, it should be emphasized that compared to the general regression-based detection module, EAST, the overall performance of our algorithm can be improved by at least 21.1% and up to 41.7%. The experimental results demonstrate that each module in Table 3 has a certain contribution to the overall performance, thus verifying the necessity of each part.

#### 4.2.4. Disadvantages and Advantages

The main disadvantage of the proposed method is not an end-to-end framework. The traditional MSER method is used in our character extraction module, and the accuracy rate is 68.3% in CTW1500 data set, while the accuracy rate of CRAFT algorithm is 75.7%. Due to unsatisfactory text detection performance of MSER, the accuracy of the proposed method is not good enough compared to the existing methods, especially CRAFT.

Although the performance of our method is slightly inferior to the CRAFT method, one of the biggest advantages of our method is that it does not depend too much on a specific data set. In order to test the generalization of the algorithm, Table 4 lists the comparison results of the generalization experiment of the two methods, in which CRAFT-IC15-20k.pth and CRAFT-MLT-25k.pth denote the CRAFT models trained on IC15 and IC13 + IC17 data sets, respectively. It can be seen from Table 4 that the model CRAFT-IC15-20k.pth performs well on the IC15 dataset, but its performances on the other four datasets have dropped a lot, and the performances of the model CRAFT-MLT-25k.pth on the IC15 and IC17-MLT data sets are also inferior to our method. In other words, the CRAFT model is strongly based on training samples, and its generalization ability is not particularly ideal. Fortunately, neither our character extraction module nor our PSO-TSA module depends on a specific dataset; consequently, they are highly applicable to any dataset. ICMN module can be readily plugged into to any regression-based text positioning module to complete the text instance detection.

## 5. Conclusions

In this paper, we have proposed a straightforward and efficient instance-aware curved text detector, which is composed of quadrilateral text proposal detection, character extraction, particle swarm optimization-based text shape approximation (PSO-TSA), and an instance-aware component merging network (ICMN). Without training or learning in advance, PSO-TSA can adjust the quadrilateral box obtained by a regression-based text detector to a polygon, thus approaches the shape of the text. ICMN can merge two adjacent or intersecting text components into one single text instance that is more complete. A large number of experiments have been carried out on five scene text detection benchmark datasets, H-mean performance achieve to 85.7%, 74.6%, 82.7%, 82.5% and 83.7% for IC15, IC17, MSRA-TD500, CTW1500 and Total-Text dataset, respectively. As the two core components of the text detector, the average running speeds of PSO-TSA and ICMN modules are 20.0 FPS and 30.9 FPS, respectively. Experimental results demonstrate that the proposed algorithm has competitive advantages in H-mean performance and execution speed.

In the future, we would like to replace the MSER extraction with more elegant character detection method to further improve the efficiency of the algorithm.

## Figures and Tables

**Figure 1 sensors-21-01945-f001:**
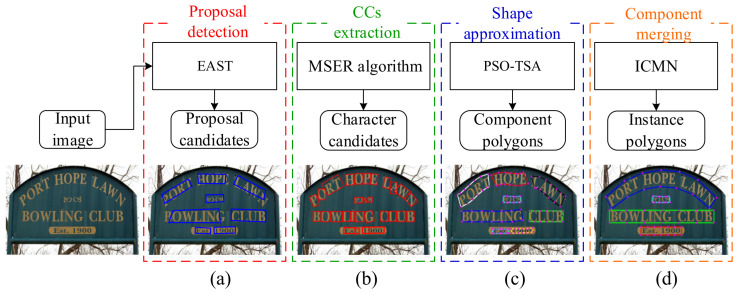
Illustration of the overall pipeline, (**a**) proposal detection module, (**b**) CCs extraction module, (**c**) shape approximation module and (**d**) component merging module.

**Figure 2 sensors-21-01945-f002:**
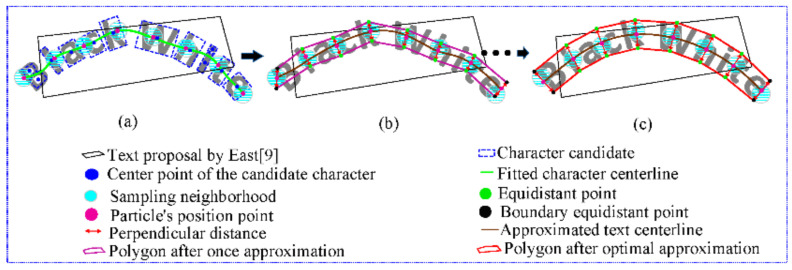
The framework of the PSO-TSA algorithm, (**a**) particle initialization process, (**b**) generating polygon by once approximation, (**c**) generating polygon by optimal approximation.

**Figure 3 sensors-21-01945-f003:**
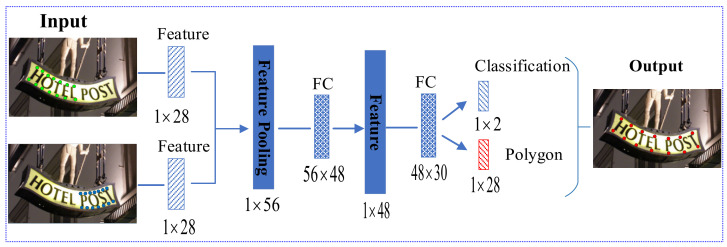
ICMN architecture.

**Figure 4 sensors-21-01945-f004:**
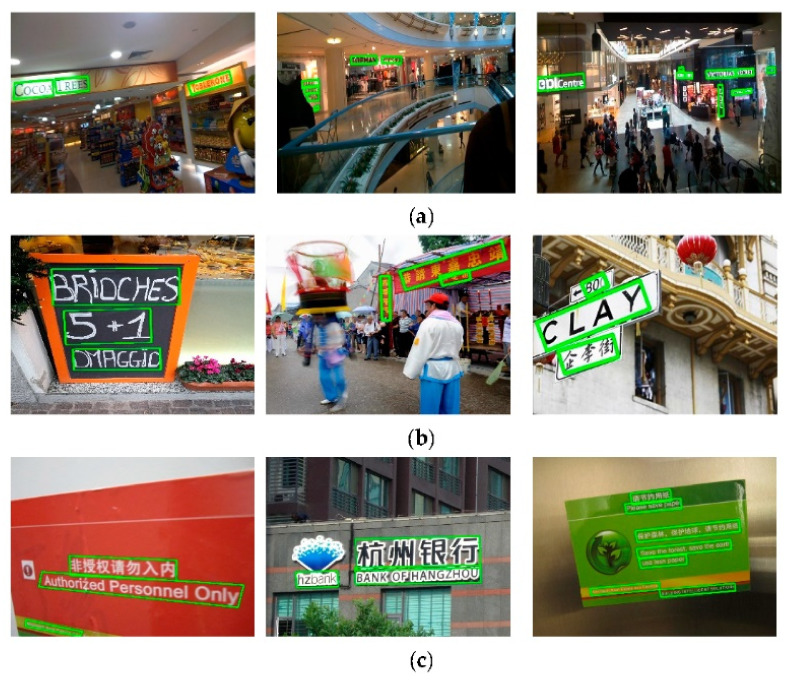
Some qualitative results of the proposed method on IC15 (**a**), IC17 (**b**), and TD500 (**c**).

**Figure 5 sensors-21-01945-f005:**
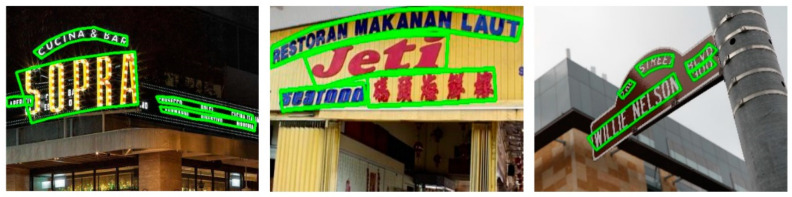
Some qualitative results of the proposed method on Total-Text.

**Figure 6 sensors-21-01945-f006:**
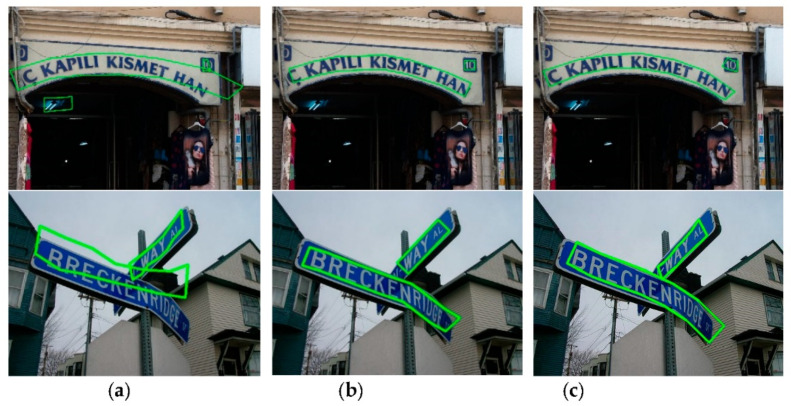
Comparison results of several methods, (**a**) CTD + TOLC [8], (**b**) CRAFT [34], and (**c**) LOMT.

**Figure 7 sensors-21-01945-f007:**
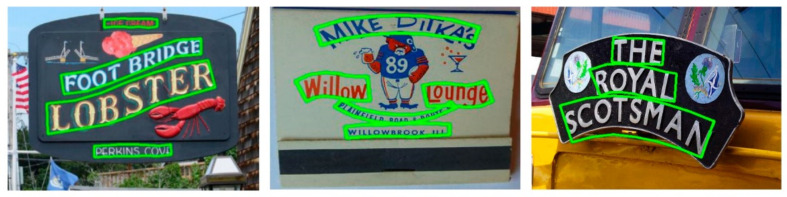
Some qualitative results of the proposed method on CTW1500.

**Table 1 sensors-21-01945-t001:** Results on quadrilateral-type datasets, such as IC15, IC17 and TD500. P, R and H refer to the precision, recall rate and H-mean, respectively. The best H score is highlighted in bold, and * denotes that the results based on multiscale tests.

Method	Backbone	Year	IC15			IC17			MSRA-TD500		
			P (%)	R (%)	H (%)	P (%)	R (%)	H (%)	P (%)	R (%)	H (%)
Zhang et al. [25]	VGG-16 + FCN	2016	71	43	54	-	-	-	83	67	74
EAST * [9]	VGG-16	2017	80.5	72.8	76.4	-	-	-	81.7	61.6	70.2
TextSnake [33]	VGG-16 + UNet	2018	84.9	80.4	82.6	-	-	-	83.2	73.9	78.3
TextBoxes++ * [5]	VGG-16	2018	87.8	78.5	82.9	-	-	-	-	-	-
PixelLink [23]	VGG-16	2018	85.5	82	83.7	-	-	-	83	73.2	77.8
Lyu et al. [32]	VGG-16 + FCN	2018	89.5	79.7	84.3	74.3	70.6	72.4	87.6	76.2	81.5
*E2E-MLT* [11]	ResNet-34 + FPN	2018	-	-	-	64.6	53.8	58.7	-	-	-
*FOTS* [18]	ResNet-50 + FPN	2018	88.8	82.0	85.3	79.5	57.5	66.7	-	-	-
PSENET-1s [17]	ResNet + FPN	2019	86.9	84.5	85.7	75.3	69.2	72.2	-	-	-
CRAFT [34]	VGG-16 + UNet	2019	89.8	84.3	**86.9**	80.6	68.2	73.9	88.2	78.2	**82.9**
FTPN [7]	ResNet + FPN	2019	68.2	78.0	72.8	-	-	-	-	-	-
ICG (SegLink++) [13]	VGG-16	2019	83.7	80.3	82.0	-	-	-	-	-	-
LOMT	ResNet-50 + FPN	2020	86.9	84.6	85.7	79.1	70.6	**74.6**	88.9	77.3	82.7

**Table 2 sensors-21-01945-t002:** Results on curved text datasets Total-Text and CTW1500. The best H score is highlighted in bold and * denotes the results based on multiscale tests.

Method	Year	Total-Text			CTW1500		
		P (%)	R (%)	H (%)	P (%)	R (%)	H (%)
EAST * [9]	2017	50.0	36.2	42.0	78.7	49.1	60.4
DeconvNet * [10]	2017	33.0	40.0	36.0	-	-	-
TextSnake [33]	2018	82.7	74.5	78.4	67.9	85.3	75.6
PSENet-1s [17]	2019	81.8	75.1	78.3	80.6	75.6	78.0
LOMO-1s [12]	2019	88.6	75.7	81.6	-	-	-
CTD + TLOC [8]	2019	77.4	69.8	73.4	77.4	69.8	73.4
ICG (SegLink++) [13]	2019	82.1	80.9	81.5	82.8	79.8	81.3
CRAFT [34]	2019	87.6	79.9	83.6	86.0	81.1	**83.5**
Wang et al. [19]	2019	80.9	76.2	78.5	80.1	80.2	80.1
Mask-TTD [29]	2020	74.5	79.1	76.7	79.7	79.0	79.4
LOMT	2020	88.2	79.6	**83.7**	85.1	79.9	82.5

**Table 3 sensors-21-01945-t003:** The ablation experiment results.

Dataset	EAST	MSER Extraction	PSO-TSA	ICMN	H (%)
CTW1500	√	X	X	X	60.4
	√	√	√	X	71.2
	√	√	√	√	81.5
Total-Text	√	X	X	X	42.0
	√	√	√	X	81.7
	√	√	√	√	83.7

**Table 4 sensors-21-01945-t004:** Comparison of generalization experimental results.

Data Sets	CRAFT-IC15-20k.pth			CRAFT-MLT-25k.pth			LOMT		
	P (%)	R (%)	H (%)	P (%)	R (%)	H (%)	P (%)	R (%)	H (%)
IC15	89.8	84.3	86.9	81.7	82.5	82.0	86.9	84.6	85.7
IC17-MLT	51.2	47.7	49.4	80.6	68.2	73.9	79.1	70.6	74.6
TD500	16.1	29.6	20.9	88.0	78.1	82.9	88.9	77.3	82.7
CTW1500	69.8	70.6	70.2	86.0	81.1	83.5	85.1	79.9	82.5
Total-Text	72.5	76.3	74.3	87.6	79.9	83.6	88.2	79.6	83.7

## Data Availability

The datasets IC15 and IC17 in this paper can be obtained from the following link: https://rrc.cvc.uab.es/?ch=4&com=downloads, accessed on 26 August 2015. The dataset TD500 in this paper can be obtained at http://www.iapr-tc11.org/mediawiki/index.php/MSRA_Text_Detection_500_Database_%28MSRA-TD500%29, accessed on 20 June 2012. The dataset CTW1500 in this paper can be obtained at https://www.github.com/Yuliang-Liu/Curve-Text-Detector, accessed on 6 December 2017. The dataset TotalText in this paper can be obtained at https://github.com/cs-chan/Total-Text-Dataset, accessed on 15 November 2017.

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
