# Peer review of "A Straightforward and Efficient Instance-Aware Curved Text Detector"

_sensors, 2021, doi:10.3390/s21061945_

Round 1

Reviewer 1 Report

The authors proposed a method for detecting curved texts in natural scene images. The idea of using PSO algorithm for detecting a polygon that covers texts seems effective and interesting. However, the accuracy of the proposed method is not good enough compared to the existing methods, especially CRAFT. Hence, more detailed consideration about the experimental results should be added. The authors should clarify the advantages and disadvantages of the proposed method compared to the existing ones using the qualitative results.

The paper is generally well-written. However, there are some points to be clarified. The definition of “text aggregation degree” in line 275 seems to be ambiguous, because if all the text pixels are completely detected, it is not difficult to detect a polygon that covers the text region. In the algorithm of ICMN, the condition of selecting two adjacent polygons should be clarified.

Reviewer 2 Report

The proposed article is interesting primarily due to the successful combination of the methods used. The authors provide an overview of the related works in this field and discuss them quite well. However, I think the article should be improved. First of all, the style. The authors use very long sentences and paragraphs, which makes it difficult to read the article. There is also a detailed discussion of papers not directly related to the proposed method. This undoubtedly overloads the text. However, the main drawback is the representation of the main algorithm. It contains parts linked to other works. The authors honestly refer to them but do not explain enough how and why they use these facts. I am not sure that the reader will be able alone to trace this connection independently. It seems to me that it makes sense to shorten the general overview and pay more attention to the presentation of the mentioned works in a new section “Background”. Proofreading would also be useful. In general, the article seems interesting and can be recommended for publication after an appropriate improvement.
